# LEARNING TRANSPORT COST FROM SUBSET CORRESPONDENCE.

**Ruishan Liu**
Department of Electrical Engineering
Stanford University

**Akshay Balsubramani**
Department of Genetics
Stanford University

**James Zou**
Department of Biomedical Data Science
Stanford University

## ABSTRACT

Learning to align multiple datasets is an important problem with many applications, and it is especially useful when we need to integrate multiple experiments or correct for confounding. Optimal transport (OT) is a principled approach to align datasets, but a key challenge in applying OT is that we need to specify a transport cost function that accurately captures how the two datasets are related. Reliable cost functions are typically not available and practitioners often resort to using hand-crafted or Euclidean cost even if it may not be appropriate. In this work, we investigate how to learn the cost function using a small amount of side information which is often available. The side information we consider captures subset correspondence—i.e. certain subsets of points in the two data sets are known to be related. For example, we may have some images labeled as cars in both datasets; or we may have a common annotated cell type in single-cell data from two batches. We develop an end-to-end optimizer (OT-SI) that differentiates through the Sinkhorn algorithm and effectively learns the suitable cost function from side information. On systematic experiments in images, marriage-matching and single-cell RNA-seq, our method substantially outperform state-of-the-art benchmarks.

## 1 INTRODUCTION

In many applications, we have multiple related datasets from different sources or domains, and learning efficient computational mappings between these datasets is an important problem (Long et al., 2017; Zamir et al., 2018). For example, we might have single-cell RNA-Seq datasets generated for the same tissue type from two different labs. Since data come from the same type of tissue, we would like to map cells between the two datasets to merge them, so that we could analyze them jointly. However, there are often complex nonlinear batch artifacts generated by the different labs. Moreover the cells are not paired—for each cell measured in the first lab, there is not an identical clone in the second lab. How to integrate or align these two datasets is therefore a challenging problem.

Optimal transport (OT) is a principled analytical framework to align heterogeneous datasets (Santambrogio, 2015). It has been increasingly applied to problems in domain adaptation and transfer learning (Seguy et al., 2017; Genevay et al., 2017; Courty et al., 2017b; Li et al., 2019). Optimal transport is an approach for taking two datasets, and computing a mapping between them in the form of a "transport plan" $\gamma$. The mapping is optimal in the sense that among all reasonable mappings (precisely defined in Section 2), it minimizes the cost of aligning the two datasets. The transport cost is given by the user and encodes expert knowledge about how datasets relate to each other. For example, if the expert believes that one data $Y$ is essentially data $X$ with added Gaussian noise, then Euclidean cost could be natural. If the cost is correctly specified, then there are powerful methods for finding the global optimal transport (Villani, 2008). A major challenge in practice, e.g. for single-cell RNA-seq, is that we and experts do not know what cost is appropriate. Users often resort to using Euclidean or other hand-crafted cost functions, which could give misleading mappings.

**Our contributions.** We propose a novel approach to automatically learn good transport costs by leveraging side information we may have about the data. In particular, aligning datasets across different conditions is a very important problem in biology, especially in single cell analysis (Butler et al., 2018), where we often have annotations that certain cluster of cells in a dataset corresponds to one cell type (e.g. B cells and T cells) based on known marker genes (Schaum et al., 2018). Then we can deduce that T cells from dataset $X$ should be at least mapped to T cells from dataset $Y$. We only need T cells to be crudely annotated in both datasets, which is reasonable; we don't need to know that a particular T cell should be mapped to another specific T cell. This gives rises to the side information that we can leverage in our algorithm — a certain subset of points in dataset $X$ should be mapped to another subset of points in dataset $Y$.

We present the first algorithm, OT-SI, to leverage subset correspondence as a general form of side information. In contrast, previous works mainly focus on pair matching problems (Li et al., 2019; Galichon & Salanié, 2010) — the extreme case of subset correspondence when the subset sizes are 1. In practice, exact one-to-one matching labels are often expensive to obtain or even intractable. OT-SI is an end-to-end framework that learns the transport cost. The intuition is to optimize over a parametrized family of transport costs, to identify the cost under which the annotated subsets are naturally mapped to each other via optimal transport. OT-SI efficiently leverages even a small amount of side information and it generalizes well to new, unannotated data. The learned transport cost is also interpretable. We demonstrate in extensive experiments across image, single-cell, marriage and synthetic datasets that our method OT-SI substantially outperform state-of-the-art methods for mapping datasets.

**Related Work** Optimal transport been well studied in the mathematics, statistics and optimization literature (Villani, 2008; Courty et al., 2017a; Li et al., 2019; Courty et al., 2017b). OT can be used to define a distance metric between distributions (e.g. Wasserstein distance) or to produce an explict mapping across two datasets. The latter is the focus of our paper. In machine learning, there has been significant work on developing fast algorithms for efficient computation of the optimal transport plan (Cuturi, 2013; Altschuler et al., 2017; Staib et al., 2017), and analyzing the properties of the transport plan under various structures and constraints on the optimization problem (Alvarez-Melis et al., 2018; Titouan et al., 2019). The previous work on learning the transport cost is done on a very different setting from ours – learning feature histogram distances between many pairs of datapoints (Cuturi & Avis, 2014). Some classical clustering (Xing et al., 2003; Bilenko et al., 2004) and alignment methods (Ham et al., 2005; Wang & Mahadevan, 2008; 2009) have realized benefits by including side information, but these semi-supervised approaches differ from our explicit parametrization and optimization of the transport cost function.

Separately, there have been recent efforts to directly map between datasets, without learning a transport cost. The standard alignment methods can be divided into two categories: GANs-based (Zhu et al., 2017; Choi et al., 2018) and OT-based (Grave et al., 2019; Alvarez-Melis et al., 2019). GAN-based approaches have been used to align single-cell RNA-seq data when pairs of cells are known to be related (Amodio & Krishnaswamy, 2018). However the exact pairing of individual cells is always not readily available or even intractable. To address this issue, our method OT-SI allows for more general correspondence between subsets, i.e., clusters, cell types and also individual cells. In the meantime, the OT-based methods always rely on Procrustes analysis (Rangarajan et al., 1997) — a linear transformation between the datasets is assumed, which lacks the flexibility to handle nonlinear artifacts and the side information cannot be utilized. In contrast, a major benefit of our approach is its graceful adaptation to partial subset correspondence information, where we frame the problem as semi-supervised.

## 2 LEARNING COST METRICS

A good choice of the cost function for optimal transport is the key to a successful mapping between two datasets. In this section, we present the algorithm OT-SI, which parametrizes the cost function with weight $\theta$ and adaptively learns $\theta$ using side information about the training data. The side information we consider is subset correspondence — a common situation when some subsets of training points are known to be related; pair matching is included as an extreme case. The learned cost function is further evaluated on the unseen test data to prove generalizability.

## 2.1 OPTIMAL TRANSPORT

Consider learning a mapping between two datasets $X = \{x^{(1)}, ..., x^{(n_X)}\}$ and $Y = \{y^{(1)}, ..., y^{(n_Y)}\}$. Here we use $n_X$ and $n_Y$ to denote the number of datapoints; each sample $x^{(i)}$ or $y^{(j)}$ could be a vector as well. We briefly recall the optimal transport framework in this setting. Given probability vectors $\mu_X$ and $\mu_Y$, the transport polytope is defined as

$$U(\mu_X, \mu_Y) := \{\gamma \in \mathbb{R}_+^{n_X \times n_Y} | \gamma \mathbf{1}_{n_X} = \mu_X, \gamma^T \mathbf{1}_{n_Y} = \mu_Y\},$$

where $\mathbf{1}_{n_X}$ ($\mathbf{1}_{n_Y}$) is the $n_X$ ($n_Y$) dimensional vector of ones. Here the probability vector $\mu_X$ ($\mu_Y$) is in the simplex $\sum_n := \{p \in \mathbb{R}_+^n : p^T \mathbf{1}_n = 1\}$ for $n = n_X$ ($n_Y$). For two random variables with distribution $\mu_X$ and $\mu_Y$, the transport polytope $U(\mu_X, \mu_Y)$ represents the set of all possible joint probabilities of the two variables. In this paper, we consider $\mu_X$ and $\mu_Y$ to represent the empirical distributions of the samples X and Y, respectively, and set $\mu = (1/n)\mathbf{1}_n$.

Given a $n_X \times n_Y$ cost matrix $C$, the classical optimal transport plan between $\mu_X$ and $\mu_Y$ is defined as $\gamma^* = \arg\min_{\gamma \in U(\mu_X, \mu_Y)} \langle \gamma, C \rangle$, where $\langle \cdot, \cdot \rangle$ denotes the Frobenius inner product. $\gamma^*$ is also called a coupling. Despite its intuitive formulation, the computation of this linear program quickly becomes prohibitive especially in the common situation when $n_X$ and $n_Y$, the sizes of the datasets, exceed a few hundred. For computational efficiency, Sinkhorn-Knopp iteration is widely used to compute the optimal transport (Cuturi, 2013). Sinkhorn-Knopp is a fast iterative algorithm for approximately solving the optimization problem with entropy regularization Santambrogio (2015):

$$\gamma^\lambda = \underset{\gamma \in U(\mu_X, \mu_Y)}{\arg\min} \langle \gamma, C \rangle - \frac{1}{\lambda} h(\gamma). \tag{1}$$

where $\lambda > 0$ is a regularization parameter and $h(\gamma) = -\sum_{i=1}^{n_X} \sum_{j=1}^{n_Y} \gamma_{ij} \log \gamma_{ij}$ denotes the entropy. The regularized solution $\gamma^\lambda$ converges to the classical one $\gamma^*$ when the regularization diminishes, i.e., $\lambda \to \infty$, with exponential convergence rate (Cominetti & San Martín, 1994). The transport $\gamma^\lambda$ treats $X$ and $Y$ symmetrically.

The cost matrix is retrieved from the cost function $C_{ij} = c(x^{(i)}, y^{(j)})$. A good choice of the cost function is the key to influencing the learned mapping $\gamma^\lambda$. However, reliable cost functions are typically not available and Euclidean cost is mostly used. In this paper, the representation of the cost function is adaptively learned using side information about the data.

## 2.2 SIDE INFORMATION

Subset correspondence describes a common situation when certain subsets of points are known to be related. For example, images with the same objects should always be mapped together in domain adaptation tasks (Courty et al., 2017b), while cells in a single-cell dataset need to be aligned to those with the same cell type, where the cell type annotation is available.

Given $m$ corresponding subsets, we write $S_X^{(k)}$ and $S_Y^{(k)}$, $k = 1, ..., m$, to denote the sets of data indices in the corresponding subsets, i.e., $\{x^{(i)} \mid i \in S_X^{(k)}\}$ and $\{y^{(i)} \mid i \in S_Y^{(k)}\}$. Note that $S_X^{(k)}$ and $S_Y^{(k)}$ could have different probability mass. If $\frac{|S_X^{(k)}|}{n_X} \leq \frac{|S_Y^{(k)}|}{n_Y}$, we take this side information to be that $S_X^{(k)}$ should be mapped into $S_Y^{(k)}$. In other words, all the other entries of the transport matrix $\gamma$ that maps $S_X^{(k)}$ to outside of $S_Y^{(k)}$ should be 0. Everything is swapped if $\frac{|S_Y^{(k)}|}{n_Y} \leq \frac{|S_X^{(k)}|}{n_X}$. Mathematically, this side information corresponds to the constraint that $\gamma_{ij} = 0$ for $(i, j) \in S_0$, where

$$S_0 = \bigcup_{k=1}^{m} \{(i,j) | i \in S_X^{(k)}, j \notin S_Y^{(k)}, \frac{|S_X^{(k)}|}{n_X} \leq \frac{|S_Y^{(k)}|}{n_Y}\} \cup \{(i,j) | i \notin S_X^{(k)}, j \in S_Y^{(k)}, \frac{|S_Y^{(k)}|}{n_Y} \leq \frac{|S_X^{(k)}|}{n_X}\} \tag{2}$$

Note that pair matching is an extreme case of subset correspondence with subset size 1 — that is, the exact pairwise relation is known. The pair matching problem has been addressed in literature (Li et al., 2019; Galichon & Salanié, 2010). However, in practice, exact one-to-one matching labels are often expensive to obtain or even intractable. In this paper, we show that subset correspondence, as a small amount of side information, can significantly aid cost function learning.

We investigate how to learn the cost function using subset correspondence information between training datasets $X_{\text{train}}$ and $Y_{\text{train}}$. The learned cost function is evaluated via the mapping quality on the test datasets $X_{\text{test}}$ and $Y_{\text{test}}$. Note that the training and test sets are not necessarily under the same distribution. We demonstrate the power of OT-SI in generalizing to new subsets that were not seen in the training process.

## 2.3 THE OT-SI ALGORITHM

Our ultimate goal is to learn a cost function $c(\cdot)$, such that *the computed optimal transport $\gamma_{\text{train}}^{\lambda}$ satisfies the side information given in Eq.* (2) *as faithfully as possible.*

**Cost function parametrization.** When the cost function $c(\cdot)$ is Euclidean, the entry of the cost matrix is computed as $C_{ij} = c(x^{(i)}, y^{(j)}) = \sum_{k=1}^{d} (x_k^{(i)} - y_k^{(j)})^2$, where $d$ is the data dimension. To learn the cost function systematically, we parametrize it as $C_{ij}(\theta) = c(x^{(i)}, y^{(j)}; \theta)$ with weight $\theta$. Here the function form $c(\cdot)$ can be chosen by users. To illustrate the improvement over the commonly-used Euclidean cost, we parametrize $c(x^{(i)}, y^{(j)}; \theta)$ as a polynomial in $(x_1^{(i)}, ..., x_{d_x}^{(i)}, y_1^{(j)}, ..., y_{d_y}^{(j)})$ with coefficients $\theta$ and degree 2 for low-dimensional data. The Euclidean cost is equivalent to a specific choice of $\theta_0$ which is set as the initialization; see Appendix for more discussions. For high-dimensional data, the memory required to store the second order polynomials becomes too large and we use a fully connected neural network to parametrize $c(x^{(i)}, y^{(j)}; \theta)$ with input $(x_1^{(i)}, ..., x_{d_x}^{(i)}, y_1^{(j)}, ..., y_{d_y}^{(j)})$ and weights $\theta$. Throughout this paper, the polynomial parametrization is used if not specified.

The optimal transport solution is characterized by $\theta$ as

$$\gamma^{\lambda}(\theta) = \underset{\gamma \in U(\mu_X, \mu_Y)}{\arg\min} \langle \gamma, C(\theta) \rangle - \frac{1}{\lambda} h(\gamma). \tag{3}$$

Then the problem can be formulated as optimizing $\theta$ to make the transport $\gamma^{\lambda}(\theta)$ approximately satisfy the conditions defined in Eq. (2), penalizing deviation of the solution from these constraints with the loss

$$L^{\lambda}(\theta) = \sum_{(i,j) \in S_{0,\text{train}}} ||\gamma_{ij,\text{train}}^{\lambda}(\theta)||_2^2. \tag{4}$$

**Theorem 1.** *For any $\lambda > 0$, the optimal transport plan $\gamma^{\lambda}(\theta)$: $\mathbb{R}^{d_\theta} \to U(\mu_X, \mu_Y)$ is $C^{\infty}$ in the interior of its domain.*

The infinite differentiability of the Sinkhorn distance is previously-known (Luise et al., 2018); Thm. 1 proves that the Sinkhorn transport plan also has this desirable property. Because Thm. 1 guarantees that $\gamma^{\lambda}(\theta)$ is infinitely differentiable, we are able to optimize $L^{\lambda}(\theta)$ in Eq. (4) by gradient descent. In practice, we iterate Sinkhorn's update a sufficient number of times to converge to $\gamma^{\lambda}$. Each iteration is a matrix operation involving the cost matrix $C(\theta)$, and when the number of iterations is fixed, we can propagate the gradient $\nabla_\theta$ through all of the iterations using the chain rule. Updating $\theta$ by one forward and backward pass has complexity of $\mathcal{O}(n^2)$ up to logarithmic terms. Hence OT-SI has the same complexity as the Procrustes-based OT methods which alternatively optimize over coupling matrix and linear transformation matrix (Grave et al., 2019; Alvarez-Melis et al., 2019). To further boost the performance, we propose to use a mimic learning method for initialization, which does not need to propagate the gradient. The pseudocode for OT-SI is in Algorithm 1 and details are in the Appendix. The proof of Thm. 1, mimic learning algorithm, details, and discussions about convergence are also in the Appendix.

## 2.4 EXPERIMENTS SETUP

The OT-SI algorithm is carried out in Pytorch (Paszke et al., 2017) and trained with GPU. The model is fitted on training set, and evaluated on test set. We use validation set for hyperparameter selection and early stopping. We evaluate OT-SI with different types of data and correspondence information.

**Comparison methods.** We use optimal transport with Euclidean cost function as a baseline for comparison, referred as "OT-baseline". We also compare our result with state-of-the-art GAN-based data alignment methods, MAGAN (Amodio & Krishnaswamy, 2018) and CycleGAN (Zhu et al.,

---

**Algorithm 1** OT-SI

---

**Require:** training datasets $X_{\text{train}}$ and $Y_{\text{train}}$, corresponding subsets index $S_{X_{\text{train}}}^{(k)}$ and $S_{Y_{train}}^{(k)}$ ($k = 1, ..., m_{\text{train}}$), step size $\alpha$, total training steps $T$, weights $\theta_0$ from initialization procedure, Sinkhorn regularization parameter $\lambda$ and number of Sinkhorn iterations $n_{\text{Sinkhorn}}$.

1: $n_X = \text{length}(X_{\text{train}})$, $n_Y = \text{length}(Y_{\text{train}})$, $\mu_X = (1/n_X)\mathbf{1}_{n_X}$, $\mu_Y = (1/n_Y)\mathbf{1}_{n_Y}$
2: Initialize $\theta = \theta_0$
3: **for** $t = 1$ to $T$ **do**
4:     Compute cost matrix $C(\theta)$ with entries $C_{ij}(\theta) = c(x^{(i)}, y^{(j)}; \theta)$.
5:     Solve $\gamma_{\text{train}}^{\lambda}(\theta) = \arg\min_{\gamma \in U(\mu_X, \mu_Y)} \langle \gamma, C(\theta) \rangle - \frac{1}{\lambda} h(\gamma)$ with Sinkhorn's update with $n_{\text{Sinkhorn}}$ iterations.
6:     Derive $\nabla_{\theta} \gamma_{\text{train}}^{\lambda}(\theta)$ by backpropagating the gradient through all Sinkhorn-Knopp iterations.
7:     Update weights $\theta := \theta - \alpha \sum_{(i,j) \in S_{0,\text{train}}} \gamma_{ij,\text{train}}^{\lambda}(\theta) \nabla_{\theta} \gamma_{ij,\text{train}}^{\lambda}(\theta)$.
8: **end for**

---

2017), as well as the OT-based methods, RIOT (Li et al., 2019) which is developed for specific pair matching applications and the Procrustes-based OT (Grave et al., 2019), referred as "OT-Procrustes". For MAGAN and CycleGAN, the matching point for a source sample is set as its nearest neighbor in the target after mapping. Among the five comparison methods, OT-baseline, OT-Procrustes and CycleGAN do not use any side information; MAGAN makes use of matching pairs; RIOT requires the one-to-one matching labels for all the datapoints. Because MAGAN and RIOT requires pairwise correspondence, they are not applied in some experiments and these are marked as N/A. We use the same settings and hyperparameters for the comparison methods as in their original implementations.

**Evaluation metrics.** When the subset correspondence is known on the test set (not shown to the algorithm), we evaluate a transport plan $\gamma$ by how much it satisfies the correspondence. Mathematically, we define *subset matching accuracy*:

$$\text{Accuracy} = \frac{\sum_{k=1}^{m} \sum_{i \in S_{X_{\text{test}}}^{(k)}} \sum_{j \in S_{Y_{\text{test}}}^{(k)}} \gamma_{ij,\text{test}}}{\sum_{k=1}^{m} \min\{|S_{X_{\text{test}}}^{(k)}|/n_{X_{\text{test}}}, |S_{Y_{\text{test}}}^{(k)}|/n_{Y_{\text{test}}}\}}.$$

From the definition, $0 \leqslant \text{Accuracy} \leqslant 1$ gives the probability of mapping to the correct corresponding subsets. When all the test datapoints are mapped into the correct subsets, the accuracy is 1; when all the data are matched to the wrong subsets, accuracy is 0. As an extreme example, pair matching is equivalent to subset correspondence with subset sizes 1, referred to as *pair matching accuracy*. In the next few sections, we thoroughly evaluate OT-SI and several state-of-the-art methods in extensive and diverse experiments—aligning single-cell RNA-seq data to correct for batch effects, aligning single-cell gene expression and protein abundances, a marriage data, an image dataset, and the synthetic twin-moon data for illustration.

## 3 BENCHMARK ON SYNTHETIC DATASETS

We first experiment with the benchmark toy example for domain adaptation — two moon datasets—to illustrate the challenges of data alignment, before we move onto complex real-world data (Germain et al., 2013; Courty et al., 2017b). The dataset is simulated with two domains, source and target. As shown in Fig. 1a, each domain contains two standard entangled moons. The two moons are associated with two different classes, denoted by circle and crossing respectively. The target (colored in orange), is built by adding noise to the source (colored in orange) and rotating by 60 degrees. In the experiment, we generate the training, test and validation datasets with 100, 100, and 50 samples of each moon. We set the parameter $\lambda = 10^3$ and the number of Sinkhorn-Knopp iterations $N = 200$. The algorithm is run for 100 epochs with step size 1. There are two types of side information available for OT tasks: (i) subset correspondence — datapoints are known to be mapped into the corresponding moon class; (ii) pair matching — known matched datapoints after rotation. The result is averaged over 10 (50) independent runs when the side information is subset correspondence (pair matching).

**Baseline performance.** The optimal transport plan under Euclidean cost function is depicted in Fig. 1b. Datapoints learned to be matched are connected by solid curves. The red curves indicate wrong

Table 1: Subset matching and pair matching accuracy on test data for two moon datasets. Here the subset (pair) matching accuracy corresponds to the proportion of the data points that are aligned to the correct moon (data points) on the test set. Higher is better. We generated 10 independent datasets and the standard deviation is shown.

|  | Side Information | OT-Baseline | OT-SI | OT-Procrustes | MAGAN | CycleGAN | RIOT |
|---|---|---|---|---|---|---|---|
| Subset Matching | Subsets | 72% | **100**% | 59% ± 14% | N/A | 48% | N/A |
|  | 1 Pair | 72% | **93**% ± 1% | 59% ± 14% | 46% ± 2% | 48% | N/A |
|  | 10 Pairs | 72% | **99.7**% ± 0.1% | 59% ± 14% | 71% ± 2% | 48% | 46.9% |
| Pair Matching | Subsets | 2% | **87**% ± 1% | 46% ± 14% | N/A | 0% | N/A |
|  | 1 Pair | 2% | **53**% ± 5% | 46% ± 14% | 0.94% ± 0.08% | 0% | N/A |
|  | 10 Pairs | 2% | **86**% ± 2% | 46% ± 14% | 1.2% ± 0.2% | 0% | 0% |

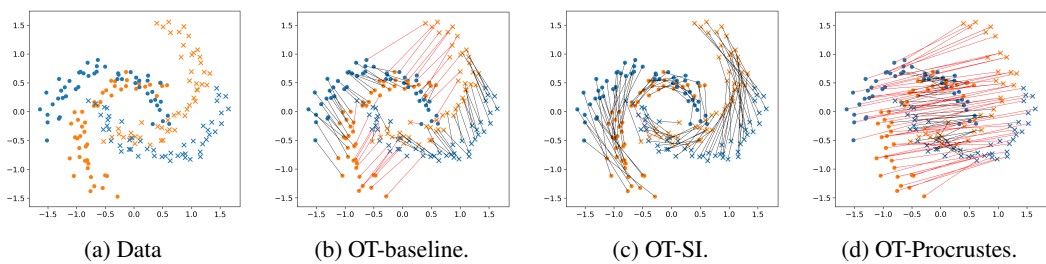

(a) Data      (b) OT-baseline.      (c) OT-SI.      (d) OT-Procrustes.

Figure 1: Illustration of the two moon datasets and the optimal transport result. The target domain (blue) is built by adding noise to the source (orange) and rotating by 60 degrees. Corresponding subsets are denoted by circles and crossings. (b-e) Optimal transport plan under (b) Euclidean cost (OT-baseline) (c) learned cost function by OT-SI and (d) Procrustes-based OT. Points learned to be matched are connected by solid curves. When a datapoint is matched to the *wrong* subset, i.e, to the other moon, the connection curve is colored by red.

transports which map the data into the wrong subset, i.e., the other moon. As shown by Fig. 1b, most wrong transports are between the points at the edges of the moons. In the euclidean space, the edge of one moon becomes "closer" to the other moon after rotation, which leads to a small cost between datapoints in different moon classes. A new cost function which captures the rotation property is expected. Quantitatively, only 72% of the data are mapped into the correct moon it belongs to and only 2% of the data are matched to their corresponding pairs, as given in Table 1.

**Subset correspondence.** We first evaluate our method OT-SI when the side information is only that the data on the corresponding moons are known to be related—i.e. $S_X^{(1)}$ and $S_X^{(2)}$ are the two moons in dataset one, and $S_Y^{(1)}$ and $S_Y^{(2)}$ are the two moons in dataset two. With the learned cost function, almost all the datapoints are mapped into the corresponding moon, i.e., the subset matching accuracy on the test data achieves 100% for both methods, as shown in Table 1. The results are averaged for 10 independent runs. Interestingly, although there is no pair matching information provided during training, the learned cost function significantly improves the matching performance. As shown in Table 1, 87% datapoints are transported into their exact matching points after rotation. The learned mapping of OT-SI is depicted in Fig. 1c. The rotation property of the datasets is correctly captured. In contrast, OT-Procrustes sometimes learns as good as OT-SI, similar to Fig. 1c, but sometimes mistakenly learns rotation as flipping, indicated by Fig. 1d. This results in overall worse accuracy and large variance for OT-Procrustes.

**Pair matching.** OT-SI demonstrates substantial improvement when the pair matching information is provided—only 1 and 10 pairs are known out of the total 100 training pairs. The matching pairs are randomly selected from the training data. OT-SI significantly outperforms all four bench methods particularly when the number of known pairs is very limited. Even when only 1 matching pair is provided, the learned cost function greatly improves the OT performance, as given in Table 1. The improvement here is largely attributable to the unlabeled data, i.e., the datapoints without any pair

Table 2: Subset matching and pair matching test accuracy for the alignment of protein and RNA expression data in CITE-seq CBMCs experiment. Here subset (pair) matching accuracy denotes the proportion of the cells that are aligned to the correct cluster (cells). Higher is better. For alignment in the original space, the expression of 5001 mRNAs are mapped to the expression of 13 proteins. For alignment in the embedding space, the first 10 principal components are used for both RNA and protein expression datasets.

| | Embedding Space (10:10) | | | Original Space (5001:13) | | |
|---|---|---|---|---|---|---|
| | OT-Baseline | OT-SI | OT-Procrustes | OT-Baseline | OT-SI | OT-Procrustes |
| Subset Matching | 9.9% | 56.1% | 44.3% | N/A | 43.9% | N/A |
| Pair Matching | 0 % | 3.2% | 1.0% | | 0.8% | |

matching information. For comparison, we carry out another experiment with only 3 unlabeled data, with all other settings unchanged. The algorithms are not able to learn the right cost function anymore — the test pair matching accuracy is only $0.5\%$ after learning, even worse than the Euclidean baseline. With 199 unlabeled data, the accuracy achieves $53\%$. In contrast, the competing methods MAGAN and RIOT learn barely any patterns, because too few labeled datapoints are available and the unlabeled ones are wasted.

## 4    BIOLOGICAL MANIFOLD ALIGNMENT

In this section, we implement our method OT-SI to learn a cost function that aligns biological manifolds with partial supervision — annotations of some cell types or clusters, which is the common situation in biological studies. The pair matching methods, MAGAN and RIOT, are not compared here because the the cell-to-cell matching information is not available. Similar to Sec. 3, the CycleGAN does not learn correctly for these data types and is not presented here. We show that OT-SI has substantial improvement for aligning datasets with different data types and aligning data from different batches.

### 4.1    ALIGNMENT OF PROTEIN AND RNA SEQUENCING DATA

How to align datasets with different data types has been a major topic in many fields. For example, in single-cell studies, RNA and protein sequencing can both be done at cellular resolution. How to map between those two types of data, i.e., map cells with certain mRNA level to cells with certain protein level, becomes critical for downstream studies such as RNA and protein co-expression analysis.

We demonstrate the power of OT-SI in learning cost function that aligns two different data types — RNA and protein expression in the CITE-seq cord blood mononuclear cells (CBMCs) experiments (Stoeckius et al., 2017). The dataset is subset to 8,005 human cells with the expression of 5,001 highly variable genes and 13 proteins. Fifteen clusters are identified using the Louvain modularity clustering. The CITE-seq technology has enabled the simultaneous measurement of RNA and protein expression at single-cell level, hence the ground truth about the cell pairing is available. To emulate the common situation, we only use the information of cluster correspondence in the training and report the performance of both subset (cluster) matching and pair matching for test. We randomly sampled 500 and 500 cells for validation and test purpose. When OT-SI is learned in the original data space with the expression of 5001 mRNA and 13 proteins for each cell, we use a fully connected neural network to parametrize the cost function, with two hidden layers of 100 and 5 neurons.

We align the RNA and protein expression datasets in two scenarios: i) the embedding space where we use the first 10 PCs for both datasets; ii) the original expression space. Table 2 shows that OT-SI substantially outperforms other methods. OT-SI is able to learn good cost function when the dimensions of the two datasets are highly unbalanced as 5001:13. The learned cost metrics in the original expression space can be used for future biological analysis on the effect and relation between RNA and protein expressions. Although the single-cell sequencing data are noisy and the algorithm is not informed of any matching pair during training, the accuracy of test pair matching is improved.

Table 3: Subset matching accuracy on the two held-out cell types, T cell and immature T cell, for aligning FACS and droplet data. The algorithms OT-SI and OT-Procrustes are trained on 2, 5 and 8 other cell types. Higher is better.

| #Training Cell Types | 2 | 5 | 8 | 0 (OT-Baseline) |
|---|---|---|---|---|
| OT-SI | 70.0% | 75.0% | 83.8% | |
| OT-Procrustes | 36.9% | 56.8% | 61.5% | 70.0% |

Table 4: Subset matching accuracy on the two held-out digits 3 and 5 when trained our model on the rest eight digits for MNIST dataset. OT is used to align the original images with the perturbed ones.

| | original | watering | swirl | sphere | flip |
|---|---|---|---|---|---|
| OT-baseline | 70.0% | 62.5% | 57.5% | 35.0% |
| OT-SI | **72.5**% | **75.0**% | **65.0**% | 57.5% |
| OT-Procrustes | 66.3% | 48.6% | 39.4% | **83.9**% |

## 4.2 BATCH ALIGNMENT

Another fascinating biological application of optimal transport is to align data from different batches. In biological studies, the samples processed or measured in different batches usually result in non-biological variations, known as batch effect (Chen et al., 2011; Haghverdi et al., 2018). Here we use OT to align two batches of data[1] — data collected with fluorescence activated cell sorting (FACS) and droplet methods. In this case, the cell types are used as the subset correspondence information. For illustration purposes, we subsample the top 10 celltypes with 400 samples of each. There are 1,682 genes after filtering and the first 10 principal components are used for analysis. The dataset is split into training, validation and test sets with ratio 50%, 20% and 30%. The experiment setting is the same as in Sec. 3.

We demonstrate the power of our learned metric in generalizing to *entirely new cell types* that were not used to train the cost. This is a hard task (a zero-shot learning task), and is more realistic because in most settings we only have partial annotations for cell types and we would like the mapping to generalize to all of the data. To do this, we choose two cell types — T cell and immature T cell — as held-out and train on the rest. Among all the ground truth expert annotations, T cell and immature T cell are most difficult to be aligned. With OT-baseline, only 70% cells are mapped into the correct cell types. Substantial improvement is achieved, as shown in Table 3. From a small number of annotated cells types, OT-SI is able to learn a transport cost that captures the batch artifacts between FACS and droplet which generalized to mapping these two new cell types. We anticipate future uses of our formulation to further investigate the cost function, particularly in biological discovery applications like isolating genes that mark single-cell heterogeneity.

## 5 EXPERIMENTS ON IMAGES AND TABULAR DATA ALIGNMENT

### 5.1 IMAGE ALIGNMENT FOR NEW DIGITS

To illustrate the use of OT-SI in image alignment, we use it to learn cost metrics in aligning images with partial annotation on the MNIST dataset which contains $28 \times 28$ images of handwritten digits. We subsample 200 images from each digit class and split them into training, validation, and test sets with ratio 50%, 20% and 30%. For illustration purposes, we use the first ten principal components for the alignment analysis. We generate four different types of perturbations to the original images, as plotted in Table 4. Then we align the original images with the perturbed ones using OT, respectively.

---

[1]https://github.com/czbiohub/tabula-muris-vignettes/blob/master/data

Table 5: Root mean square error (RMSE) and mean absolute error (MAE) of pair matching algorithms for marriage-matching dataset. Lower is better.

|      | Random | PMF  | SVD   | itemKNN | RIOT | FM  | OT-SI |
|------|--------|------|-------|---------|------|-----|-------|
| RMSE | 54.7   | 77.8 | 109.0 | 2.4     | 2.4  | 9.5 | 2.4   |
| MAE  | 36.5   | 36.1 | 62.0  | 1.6     | 1.5  | 7.5 | 1.5   |

The experiment setting is the same as in Sec. 3. Similar to Sec. 4, we test how well our algorithm generalizes to *new classes* that were not used in learning the cost function. In the experiment, we hold out digits 3 and 5, and demonstrate the metric learned on the other eight digits can help the alignment of digits 3 and 5. We achieve consistent improvement over the baseline on all four distribution types, as indicated by Table 4.

## 5.2   MARRIAGE DATASET FOR PAIR MATCHING

While OT-SI is designed for the more general form of side information — subset correspondence, it can also be used for pair matching purpose. We finally benchmark it from the comparison with other state-of-the-art *pair matching* methods, including RIOT, factorization machine model (FM) (Rendle, 2012), probabilistic matrix factorization model (PMF) (Mnih & Salakhutdinov, 2008), item-based collaborative filtering model (itemKNN) (Cremonesi et al., 2010), classical SVD model (Koren et al., 2009) and baseline random predictor model. These methods were also used as comparisons in Li et al. (2019). We follow the same experimental protocol as in Li et al. (2019) for the Dutch Household Survey (DHS) dataset. The exact matching matrix between 50 datapoints with 11 features are known. We note that the coupling matrix is treated as continuous and OT-Procrustes is not applicable.

The performance is evaluated by the root mean square error (RMSE) and the mean absolute error (MAE) for the predicted matching matrix, as given in Table 5. When used for pair matching purposes, OT-SI report comparable performance to state-of-the-art matching algorithms. Note that this marriage dataset was a primary motivating dataset used to design RIOT (Li et al. (2019)), and therefore we expect RIOT to perform very well for this task.

## 6   DISCUSSION

In this paper, we study the problem of learning the transport cost using side information in the form of a small number of corresponding subsets between the two datasets. This is a new problem formulation, to the best of our knowledge. Previous works rely on more restricted information such as that specific pairs of points should be aligned. In settings such as genomics and images, it is often difficult to say that a single point in dataset one should be mapped onto a particular point in dataset two. It is more common to have partial annotation of subsets of points—e.g. T cells are annotated in two single-cell RNA-seq datasets—which motivates our generalization.

We propose a flexible and principled method to learn the transport cost with side information. Experiments demonstrate that they work significantly better than state-of-the-art methods when the side-information is very limited, which is often the case. We compare against state-of-the-art methods for the special case when the side information consists of matching pairs, since we are not aware of other published OT methods that deal with the more general subset correspondence. One interesting reason for the improved performance is that by learning the transport cost directly, our algorithms are more efficiently using all of the unannotated datapoints that are not in any pairs or subsets. These unannotated data act as regularization (similar to in semi-supervised 1learning), which enables the model to avoid overfitting to the limited side information. An interesting direction of future work is to interpret the learned cost function for insights on how the datasets differ.

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

## A   COST FUNCTION

In Sec. 2.3, the cost function is parametrized as $C_{ij}(\theta) = c(x^{(i)}, y^{(j)}; \theta)$ with weight $\theta$. For low-dimensional data, we choose $c(x^{(i)}, y^{(j)}; \theta)$ as a polynomial in $(x_1^{(i)}, ..., x_{d_x}^{(i)}, y_1^{(j)}, ..., y_{d_y}^{(j)})$ with coefficients $\theta$ and degree 2:

$$C_{ij}(\theta) = \sum_{\substack{\alpha,\beta \in \mathbb{Z}^d, \, \alpha,\beta \geqslant 0 \\ 1 \leqslant \mathbf{1}_d^T \alpha + \mathbf{1}_d^T \beta \leqslant 2}} \theta_{\alpha,\beta} \prod_{k=1}^{d_x} \prod_{l=1}^{d_y} (x_k^{(i)})^{\alpha_k} (y_l^{(j)})^{\beta_l}. \tag{5}$$

Here we do not require the dimension of the two datasets $d_x$ and $d_y$ to be the same. When the two datasets $X$ and $Y$ have different features, the learned weights indicate the coupling between different features in the data mapping. In general, the function form $c(\cdot)$ can be chosen by users. We have also investigated parametrizing $C(\theta)$ as a small fully connected neural network and achieved very similar performance.

The Euclidean cost is equivalent to a specific choice of $\theta$, when the two datasets have the same feature space $d_x = d_y = d$. The entry of the cost matrix is computed as $C_{ij} = c(x^{(i)}, y^{(j)}; \theta) = \sum_{k=1}^d ((x_k^{(i)})^2 + (y_k^{(j)})^2 - 2x_k^{(i)} y_k^{(j)})$.

## B   OT-SI ALGORITHM

The Lagrangian dual of Eq. (3) is

$$\max_{u,v} \ \mu_X^T u + \mu_Y^T v - \frac{1}{\lambda} \sum_{i=1}^{n_X} \sum_{j=1}^{n_Y} e^{-\lambda(C_{ij}(\theta) - u_i - v_j)} \tag{6}$$

By Sinkhorn's scaling theorem (Sinkhorn & Knopp, 1967), the optimal transport plan $\gamma^\lambda(\theta)$ is computed as

$$\gamma^\lambda(\theta) = \operatorname{diag}(e^{\lambda u^*(\theta)}) e^{-\lambda C(\theta)} \operatorname{diag}(e^{\lambda v^*(\theta)}), \tag{7}$$

where $u^*(\theta)$ and $v^*(\theta)$ are the solutions to the dual problem in Eq. (6).

### B.1   PROOF OF THEOREM 1

The analysis for Theorem 1 follows the strategy of the proofs of Theorem 2 in (Luise et al., 2018).

**Theorem 1.**   *For any $\lambda > 0$, the optimal transport plan $\gamma^\lambda(\theta)$: $\mathbb{R}^{d_\theta} \to U(\mu_X, \mu_Y)$ is $C^\infty$ in the interior of its domain.*

*Proof.* Based on Eq. (7), the optimal transport plan $\gamma^\lambda(\theta)$ is a smooth function when $u^*(\theta)$, $v^*(\theta)$ and $C(\theta)$ are smooth. In the meantime, the cost $C(\theta)$ is a linear function of $\theta$, as indicated by Eq. (5). Thus to prove the smoothness of $\gamma^\lambda(\theta)$, we only need to demonstrate $u^*(\theta)$ and $v^*(\theta)$ are both smooth in $\theta$.

Here we define

$$\sigma(\theta; u, v) = -\mu_X^T u - \mu_Y^T v + \frac{1}{\lambda} \sum_{i=1}^{n_X} \sum_{j=1}^{n_Y} e^{-\lambda(C_{ij}(\theta) - u_i - v_j)}$$

The dual problem in Eq. (6) becomes $\min_{u,v} \sigma(\theta; u, v)$. From the definition, $\sigma(\theta; u, v)$ is smooth and strictly convex in $(u, v)$. Note that $C(\theta)$ is linear in $\theta$. Then for any fixed $\theta$ in the interior of $\mathbb{R}^{d_\theta}$, there exits $(u^*(\theta), v^*(\theta))$ such that $\sigma(\theta; u^*(\theta), v^*(\theta)) = \min_{u,v} \sigma(\theta; u, v)$. The function $\nabla_{(u,v)} \sigma(\theta; u, v) \in C^\infty$ due to the smoothness of $\sigma(\theta; u, v)$. Now we fix $(u_0, v_0, x_0)$ such that $\nabla_{(u,v)} \sigma(\theta_0; u_0, v_0) = 0$. The strict convexity of $\sigma(\theta; u, v)$ ensures that $\nabla_{(u,v)}^2 \sigma(\theta_0; u_0, v_0)$ is invertible.

From implicit function theorem, we can always find a function $f$ and a subset $U_{\theta_0} \in \mathbb{R}^{d_\theta}$ such that i) $f(\theta_0) = (u_0, v_0)$; ii) $\nabla_{(u,v)} \sigma(\theta; f(\theta)) = 0$ for any $\theta \in U_{\theta_0}$; iii) $f \in C^\infty(U_{\theta_0})$. That is, $f(\theta)$ is a

stationary point of the function $\sigma$ for any $\theta$ in $U_{\theta_0}$. Together with the strict convexity of $\sigma$, we derive $f(\theta) = (u^*(\theta), v^*(\theta))$. Recalling $f \in C^\infty(U_{\theta_0})$, we prove that $(u^*(\theta), v^*(\theta))$ is $C^\infty$ in the interior of its domain.

## B.2 Convergence properties

The gradient of $\gamma^\lambda(\theta)$ in Eq. (7) is computed as

$$\nabla_\theta \gamma_{ij}^\lambda(\theta) = \lambda \gamma_{ij}^\lambda(\theta) \left( \nabla_\theta u_i^*(\theta) + \nabla_\theta v_j^*(\theta) - \nabla_\theta C_{ij}(\theta) \right)$$

The convergence of scaling factors $e^{\lambda u(\theta)}$ and $e^{\lambda v(\theta)}$ is linear (i.e. exponential in $n_{\text{Sinkhorn}}$), with the bounded rate given by (Franklin & Lorenz, 1989); our experiments use a linear $C(\theta)$, and we do not find Sinkhorn to bottleneck convergence. Besides this point, convergence is determined by the loss landscape of $C$. Analyzing convergence to the global optimum, and the role of primal regularization of $\gamma$, is a relevant open question.

## B.3 Computing Gradient

In practice, we iterate Sinkhorn's update $N$ times to converge to $\gamma^\lambda$. More specifically, the dual solutions $u^*(\theta)$ and $v^*(\theta)$ for Eq. (6) are computed by iterating for $N$ times

$$u \leftarrow \mu_X / e^{-\lambda C(\theta)} v$$

$$v \leftarrow \mu_Y / (e^{-\lambda C(\theta)})' u$$

For the optimal plan $\gamma^\lambda(\theta)$ given in Eq. (7), its gradient $\nabla_\theta \gamma^\lambda(\theta)$ is obtained by propagating through all the iterations using the chain rule and implemented by Pytorch. Thus updating $\theta$ by one forward and backward pass has complexity of $\mathcal{O}(n^2)$ up to logarithmic terms. OT-SI has the same complexity as the Procrustes-based OT methods which alternatively optimize over coupling matrix and linear transformation matrix. Taking the image alignment for watering MNIST dataset as an example, the running time for OT-SI and OT-Procrustes is 229.5s and 196.7s, respectively.

## C Mimic Learning for Initialization

In this section, we derive a mimic learning as an fast initialization method to boost the performance and accelerate the learning. While Algorithm 1 requires to differentiate through Sinkhorn updates, the mimic learning approach does not need to propagate the gradient through all the iterations and is applicable for any kind of OT algorithm. Here we take the classical optimal transport plan $\gamma^* = \arg\min_{\gamma \in U(\mu_X, \mu_Y)} \langle \gamma, C \rangle$ as an example.

As discussed in Sec. 2.3, our ultimate goal is to learn a cost function, such that the optimal transport $\gamma^*$ satisfies the side information defined in Eq. (2) as faithfully as possible. From another perspective, we force an additional set of constraints on the transport plan $\gamma$ to fulfill the condition in Eq. (2):

$$U^c = \bigcap_{k=1}^m \{\gamma | \gamma_{ij} = 0, i \in S_X^{(k)}, j \notin S_Y^{(k)}, \frac{|S_X^{(k)}|}{n_X} \le \frac{|S_Y^{(k)}|}{n_Y}\}$$

$$\cap \{\gamma | \gamma_{ij} = 0, i \notin S_X^{(k)}, j \in S_Y^{(k)}, \frac{|S_Y^{(k)}|}{n_Y} \le \frac{|S_X^{(k)}|}{n_X}\} \tag{8}$$

To quantify how much the learned $\gamma^*$ in Eq. (1) follows the side information, we compare it with

$$\hat{\gamma} = \arg\min_{\gamma \in U(\mu_X, \mu_Y) \cap U^c} \langle \gamma, C \rangle. \tag{9}$$

Here $\hat{\gamma}$ is interpreted as the optimal transport plan when the side information is completely satisfied, and $\langle \hat{\gamma}, C \rangle$ is the smallest transport distance under the constraint. With the cost function parametrized as $C_{ij}(\theta) = c(x^{(i)}, y^{(j)}; \theta)$, The optimal transport solution in Eq. (9) is characterized by $\theta$ as $\hat{\gamma}(\theta) = \arg\min_{\gamma \in U(\mu, \nu) \cap U^c} \langle \gamma, C(\theta) \rangle$.

---

**Algorithm 2** Mimic Learning for Initialization

---

**Require:** training datasets $X_{\text{train}}$ and $Y_{\text{train}}$, corresponding subsets index $S_{X_{\text{train}}}^{(k)}$ and $S_{Y_{train}}^{(k)}$ ($k = 1, ..., m_{\text{train}}$), step size $\alpha$, total steps $T_{\text{init}}$, optimal transport solver OTSolver.
1: $n_X = \text{length}(X_{\text{train}})$, $n_Y = \text{length}(Y_{\text{train}})$, $\mu_X = (1/n_X)\mathbf{1}_{n_X}$, $\mu_Y = (1/n_Y)\mathbf{1}_{n_Y}$
2: Define transport polytopes $U^* = U(\mu_X, \mu_Y)$ and constraints $\hat{U}^c$ from Eq. (8).
3: Initialize $\theta$ such that the cost function is equivalent to the Euclidean cost.
4: **for** $t = 1$ to $T_{\text{init}}$ **do**
5: $\quad$ Compute cost matrix $C$ as $C_{ij} = c(x^{(i)}, y^{(j)}; \theta)$
6: $\quad$ $\gamma^* = \text{OTSolver}(C, U^*)$, $\hat{\gamma} = \text{OTSolver}(C, U(\mu_X, \mu_Y) \cap U^c)$
7: $\quad$ $L_{\text{init}}(\theta) = \frac{\langle \hat{\gamma}, C(\theta) \rangle - \langle \gamma^*, C(\theta) \rangle}{\langle \bar{\gamma}, C(\theta) \rangle - \langle \gamma^*, C(\theta) \rangle}$
8: $\quad$ Update weights $\theta := \theta - \alpha \nabla_\theta L_{\text{init}}(\theta)$
9: **end for**

---

Then we also expect a good cost function to make the distance under constraint $\langle \hat{\gamma}, C \rangle$ to be as small as the lowest one $\langle \gamma^*, C \rangle$ as possible — optimize $\theta$ to minimize the loss

$$L_{\text{init}}(\theta) = \langle \hat{\gamma}(\theta), C(\theta) \rangle - \langle \gamma^*(\theta), C(\theta) \rangle. \tag{10}$$

We refer to this method as mimic learning, because its objective is to make the $\hat{\gamma}$ mimic the cost performance of $\gamma^*$.

Note that $\gamma^*(\theta)$ is the optimal solution for any transport matrix in $U(\mu_X, \mu_Y)$. That is, the optimal distance $\langle \gamma^*(\theta), C(\theta) \rangle \leq \langle \gamma, C(\theta) \rangle$ for any $\gamma \in U(\mu_X, \mu_Y)$. The equality holds true only when $\gamma = \gamma^*(\theta)$ for the convex transport problem. In the meantime, we have $\hat{\gamma}(\theta) \in U(\mu_X, \mu_Y) \cap U^c$. Thus the loss $L_{\text{init}}(\theta)$ is always larger or equal to 0. When zero loss is achieved, we have $\gamma^*(\theta) = \hat{\gamma}(\theta)$, coinciding with the optimal solution for $L(\theta)$ in Eq. (3).

Equation (10) describes the absolute difference between the two transport distances, but a relative difference is more desirable in practice to adjust for the scale of the objective function around $\langle \gamma^*, C \rangle$. For example, scaling the cost matrix $C(\theta)$ by a constant does not change the solutions $\gamma^*$ and $\hat{\gamma}$, but does scale the loss defined in Eq. (10) by the same constant. We modify the loss to be invariant to such scaling:

$$L_{\text{init}}(\theta) = \frac{\langle \hat{\gamma}(\theta), C(\theta) \rangle - \langle \gamma^*(\theta), C(\theta) \rangle}{\langle \bar{\gamma}, C(\theta) \rangle - \langle \gamma^*(\theta), C(\theta) \rangle}, \tag{11}$$

Here $\bar{\gamma}$ is a uniform $n_X \times n_Y$ matrix used to stand for the averaged performance of random transport plans. Eq. (11) captures how close the distance under constraint $\langle \gamma^*, C \rangle$ is to the best one, compared to other random transports.

The mimic learning is approximately solved by alternating minimization. As described in Algorithm 2, we iterate over two steps: (i) compute the value of $\gamma^*$ and $\hat{\gamma}$ while fixing $\theta$; (ii) take one gradient step with respect to $L_{\text{init}}(\theta)$ with fixed $\gamma^*$ and $\hat{\gamma}$. The computation for optimal transport plans and the optimization of $\theta$ are carried out in alternating fashion.

The OT solver is used only to estimate the value of $\gamma^*$ and $\hat{\gamma}$ in the first step, requiring no gradient propagation. Given such estimates of transport mappings $\gamma^*$ and $\hat{\gamma}$, the second step can be interpreted as learning a cost function which equates their transport costs, i.e., makes the behavior of $\hat{\gamma}$ mimic that of $\gamma^*$. In the experiments, we set $\alpha = 1$ and $T_{\text{init}} = 10$ for initialization purpose.

