# OpenReview forum: "Learning transport cost from subset correspondence"
_ICLR.cc/2020/Conference — Accept (Poster)_

### Official Review · AnonReviewer3 · 2019-10-23
**Official Blind Review #3**

**Rating:** 3

**Review:**

The authors consider the alignment problem for multiple datasets with side information via entropic optimal transport (Sinkhorn). The authors formulate it as a transport cost learning in optimal transport framework with constraints giving by side information. Empirically, the authors illustrated the effectiveness of the proposed approach over state-of-the-art on several applications (e.g. single-cell RNA-seq, marriage-matching, MNIST with its perturbed one.

The paper show good results of the proposed method for dataset alignment on several applications. However, the motivation of the dataset alignment seems quite weak (e.g. it is unclear the advantage for jointly analysis when one merge datasets by alignment). The side information is not clearly easy to obtain in practice, since it simply enforces the constraints on the transportation polytope (i.e. how one knows that which subset of a given class C_A on dataset A is matched with a certain subset of another class C_B on dataset B, it seems easy to obtain the side information about matching over the whole class C_A with C_B, and it seems nontrivial to obtain the subset matching). Additionally, some important information about the proposed method is not in the main text but in the appendix which makes the paper hard to read without checking appendix constantly (especially the main proposed method). Overall, I lean on the negative side.

Below are some of my concerns:

1) As stated above, I am curious how one can obtain the side information about the subset matching? For example, in Figure 1, it seems trivial to have a side information about the whole class matching and quite nontrivial to obtain a matching for subsets of any given class.

2) The side information enforces the constraints on the feasible set of the transportation polytope. It seems to me that one can solve the variant of optimal transport (or Sinkhorn) with those extra constraints directly, it is unclear to me how this kind of side information can navigate the transport cost learning. Moreover, given a new pairs of input, the proposed method needs to optimize the transport cost again which make the benefits to learn to transport cost quite limited. Overall, a trivial baseline for the proposed method is to solve directly optimal transport with those extra constrains (from side information) directly (maybe with the Euclidean cost metric), but not the optimal transport without extra constraints as in the current setting.

3) The authors may need to consider to reorganize the presentation. Much important information of the proposed method is not in the main text, but in the appendix which make the paper hard to follow without checking the appendix constantly. The derivation of gradient in line 6 of Algorithm 1 is not trivial. It is better if the authors give more detail information and discussion. I am quite confused about the idea to back propagate the "gradient" through Sinkhorn-Knopp iterations?

4) It seems that the complexity of gradient update of Algorithm 1 is quite high. The authors should give a comparison about time consumption in the experiments (besides the accuracy matching)

5) It seems that there is nothing in the code sharing folder?


**Experience Assessment:**

I have published one or two papers in this area.

**Review Assessment: Checking Correctness Of Derivations And Theory:**

I assessed the sensibility of the derivations and theory.

**Review Assessment: Checking Correctness Of Experiments:**

I assessed the sensibility of the experiments.

**Review Assessment: Thoroughness In Paper Reading:**

I read the paper at least twice and used my best judgement in assessing the paper.

---

> ### Author Response · Authors · 2019-11-15
> **Response**
>
> Thank you for your thoughtful review and suggestions.
>
> Regarding your motivation for dataset alignment, aligning datasets across different conditions is a very important problem in biology, especially in single cell analysis. This is highlighted in several recent high-profile papers, e.g. Butler et al Nature Biotechnology 2018 “Integrating single-cell transcriptomic data across different conditions, technologies, and species”. In these experiments, there is typically insufficient data or strong batch effects within a single dataset, and how to align datasets in order to increase sample and remove batch effect is considered one of the most important open challenges in computational genomics [Haghverdi et al Nature Biotechnology 2018]. This motivates our work on data alignment.
>
> Regarding your specific suggestions.
>
> 1. In single cell analysis, which is our motivating example, we often have annotations that certain cluster of cells in a dataset corresponds to one cell type (e.g. B cells and T cells) based on known marker genes (e.g. Tabula Muris Nature 2018). When we have datasets D1 and D2, we typically know which are the B cells in D1 and in D2 respectively. This naturally give rise to the subset matching information that we can leverage in our algorithm. We have added more discussion of this context to the text.
>
> 2. The main advantage of our approach is that we learn a good cost function for the optimal transport, which could be quite different from the Euclidean cost. With the new cost function, we can then immediately compute transport maps between new data. This allows our approach to robustly generalize to new data. The baseline experiment that you suggested has the same performance as OT-baseline in the paper, as no side information about the test data is known to the algorithms.
>
> 3. Thanks for this suggestion, we have added details about Algorithm 1 and how we backpropagate through the Sinkhorn-Knopp iterations in the revised text; see B.3 of the appendix.
>
> 4. The complexity of Algorithm 1. Updating the weight by one forward and backward pass has complexity of O(n^2) up to logarithmic terms. OT-SI has the same complexity as the Procrustes-based OT methods which alternatively optimize over coupling matrix and linear transformation matrix (Page 4). In practice, taking the MNIST watering image alignment as an example, the running time for OT-SI and OT-Procrustes is 229.5s and 196.7s, respectively.
>
> 5. The code is now in the folder.

---

### Official Review · AnonReviewer1 · 2019-10-23
**Official Blind Review #1**

**Rating:** 6

**Review:**

# Summary
This paper proposes a new way to learn the optimal transport (OT) cost between two datasets that can utilize subset correspondence information. The main idea is to use a neural network to define a transport cost between two examples instead of using the popular Euclidean distance and have a L2 regularizer that encourages the resulting transport plan to be small if two examples belong to the same subset. The results on synthetic datasets (2D, MNIST image) and real-world datasets (RNA sequence) show that the learned transport cost outperforms the Euclidean distance and Procrustes-based OT.

# Originality
- This paper considers a new problem which is to learn OT given "subset correspondence" information. The prior work does not have a mechanism to utilize such information or requires more side information such as pair-wise matching.

# Quality
- The idea of parameterizing the transport cost using a neural network is novel to my knowledge. The idea of regularizing the transport cost based on the subset correspondence information sounds sensible.
- The empirical result is quite comprehensive and convincing in that it includes from illustrative examples to real-world datasets (RNAs) and includes many relevant baselines.

# Clarity
- The paper is well-organized and well-written.
- Since this paper considers a new problem (learning OT with subset correspondence data), it would be great to motivate and explain the problem better in the introduction. RNA-seq examples in the introduction were not easy to parse as I do not have any background on biology. It would be better if the paper presented more illustrative/intuitive examples ideally with a figure that describes the problem.
- "an principled" -> "a principled"

# Significance
- This paper seems to present a neat and sensible idea for learning OT with neural networks given subset correspondence information. The problem and method are new, and the empirical results look comprehensive. One minor issue is that it is unclear how significant this new problem setup is (learning OT cost with subset correspondence). Although the paper presents real-world examples (RNAs), the paper could be stronger if it presented more real-world examples/results and emphasized that this is an important problem to solve.

**Experience Assessment:**

I have read many papers in this area.

**Review Assessment: Checking Correctness Of Derivations And Theory:**

I did not assess the derivations or theory.

**Review Assessment: Checking Correctness Of Experiments:**

I carefully checked the experiments.

**Review Assessment: Thoroughness In Paper Reading:**

I read the paper at least twice and used my best judgement in assessing the paper.

---

> ### Author Response · Authors · 2019-11-15
> **Response**
>
> Thank you for your careful review! Following your suggestion, we have added additional motivation for aligning datasets with subset correspondence. In particular, aligning datasets across different conditions is a very important problem in biology, especially in single cell analysis. This is highlighted in several recent high-profile papers, e.g. Butler et al Nature Biotechnology 2018 “Integrating single-cell transcriptomic data across different conditions, technologies, and species”. In these experiments, there is typically insufficient data or strong batch effects within a single dataset, and how to align datasets in order to increase sample and remove batch effect is considered one of the most important open challenges in computational genomics [Haghverdi et al Nature Biotechnology 2018].

---

### Official Review · AnonReviewer2 · 2019-10-24
**Official Blind Review #2**

**Rating:** 8

**Review:**

Summary:
The paper presents a gradient-based method for learning the cost function for optimal transport, applied to dataset alignment. The algorithm is based on the Sinkhorn-Knopp iterative procedure for approximating the optimal transport plan. The current paper proves that the Sinkhorn transport plan with a parameterized cost function is infinitely differentiable. The cost function can thus be optimized by unrolling the Sinkhorn iteration for a fixed number of steps and then backpropagating through the iterative updates. The cost function is optimized via a loss function derived from side information, specifically subsets of the two datasets to be aligned that contain elements that should be matched in the optimal transport plan. Experiments on a variety of synthetic and real datasets show that the proposed method is often better than, and almost always competitive with, other modern approaches.

I recommend that the paper be accepted. The paper presents a well-justified and seemingly novel approach to solving an important problem, and demonstrates empirically that it compares favorably to other approaches. The proposed technique seems to be a fairly small extension to existing work, but together with its analysis and experiments it is sufficiently novel.

Details / Questions:
* An interesting point of comparison would be methods for semi-supervised metric learning that are not necessarily tailored to the OT setting. E.g. [1] (picked fairly arbitrarily)

* Is it possible to adapt algorithms that need pairwise matches to your subset matching setting by e.g. choosing pairs in the matching subsets at random and calling them matching pairs?

* Do you have any intuitions for the pattern of results in Table 4? Why might flips be a difficult transforation for OT-SI to learn?

Trivia:
* Page 6 is blank when I print the paper. Not sure if anyone else has this problem.

References:
[1] Bilenko, M., Basu, S., & Mooney, R. J. (2004, July). Integrating constraints and metric learning in semi-supervised clustering. In Proceedings of the twenty-first international conference on Machine learning (p. 11). ACM.

**Experience Assessment:**

I do not know much about this area.

**Review Assessment: Checking Correctness Of Derivations And Theory:**

I assessed the sensibility of the derivations and theory.

**Review Assessment: Checking Correctness Of Experiments:**

I assessed the sensibility of the experiments.

**Review Assessment: Thoroughness In Paper Reading:**

I read the paper at least twice and used my best judgement in assessing the paper.

---

> ### Author Response · Authors · 2019-11-15
> **Response**
>
> Thank you for your review and your helpful suggestions!
>
> * We have added a discussion of [1] and related works on semi-supervised metric learning to the text.
>
> * This is an interesting idea to try to convert subset matches into random pairwise matches. It doesn’t work well in our experiments. This is likely because there are many ways to generate random pairs from matched subsets.
>
> * For flips, OT-Procrustes performs better because it assumes a linear transformation between datasets and has no model mismatch, but OT-SI has a more complicated parametrization. If we increase the training data size from 200 to 1000 images per class, the accuracy of OT-SI increases from 57.5% to 80.0%, while the accuracy of OT-Procrustes remains as 83.5%.
> When the perturbation is not simply linear (e.g. the other three distribution types), OT-Procrustes can behave even worse than the baseline due to the lack of flexibility to handle nonlinear artifacts.

---

### Decision · Program_Chairs · 2019-12-19

**Decision:**

Accept (Poster)

**Comment:**

The paper proposes an algorithm for learning a transport cost function that accurately captures how two datasets are related by leveraging side information such as a subset of correctly labeled points. The reviewers believe that this is an interesting and novel idea. There were several questions and comments, which the authors adequately addressed. I recommend that the paper be accepted.